# Advances in Green Synthesis of Metal Oxide Nanoparticles by Marine Algae for Wastewater Treatment by Adsorption and Photocatalysis Techniques

Ahmed E. Alprol [1], Abdallah Tageldein Mansour [2,3,*] , Abdelwahab M. Abdelwahab [2,4] and Mohamed Ashour [1,*]

[1]  National Institute of Oceanography and Fisheries (NIOF), Cairo 11516, Egypt; ah831992@gmail.com
[2]  Animal and Fish Production Department, College of Agricultural and Food Sciences, King Faisal University, Al Hofuf 31982, Saudi Arabia; amsaid@kfu.edu.sa
[3]  Fish and Animal Production Department, Faculty of Agriculture (Saba Basha), Alexandria University, Alexandria 21531, Egypt
[4]  Department of Animal Production, Faculty of Agriculture, Fayoum University, Fayoum 63514, Egypt
*   Correspondence: amansour@kfu.edu.sa (A.T.M.); microalgae_egypt@yahoo.com (M.A.)

**Abstract:** The use of algae-based green synthesis of metal oxide nanoparticles (MONPs) for bioremediation is an environmentally friendly and cost-effective alternative to conventional approaches. Algal-mediated synthesis offers several benefits over other biogenic processes, such as plants, bacteria, and fungi, including ease of synthesis, scalability, and rapid synthesis. Algae are readily available in nature, nontoxic, and can produce various types of metal oxide nanoparticles. This approach could significantly accelerate the development of novel algae-nanomaterials with improved properties and performance, leading to more efficient and cost-effective bioremediation of pollutants from water solutions, seawater, and industrial effluent. This review focuses on the biogenic fabrication of metal oxide nanoparticles based on aquatic plants (microalgae and seaweeds) due to their many advantages and attractive applications in pollutant remediation from aqueous solutions. Additionally, photocatalysis is highlighted as a promising tool for the remediation of industrial effluents due to its efficacy, ease of use, quick oxidation, cost-effectiveness, and reduced synthesis of harmful byproducts.

**Keywords:** microalgae; seaweed; nanotechnology; MONPs; green synthesis; green chemistry; organic pollutants; adsorption; photocatalysis

## 1. Introduction

The Sustainable Development Goals (SDGs) suggested by the United Nations for 2030 include a specific goal centered on sanitation and water, which has been further broken down into eight targets. One of these targets is dedicated to successful wastewater management. The SDG report has identified four essential actions that must be taken to achieve this goal [1]. These actions include reducing contamination at the source, treating contaminants, recycling treated wastewater, and recovering by-products [2]. "Green synthesis" has become a popular technique to achieve the SDGs goals by 2030 and describe the biological techniques used to create nanomaterials [3–6]. In recent times, scientists have worked to develop simple, reliable, and environmentally friendly processes for producing nanomaterials [7,8]. Various biological sources, such as bacteria, yeast, viruses, actinomycetes, fungi, seaweeds, microalgae, plant extracts, aquaculture wastes, horticultural food materials, and disposed food products, have been used to create stable and well-functionalized nanomaterials [9,10]. The biological synthesis technique has recently attracted a great deal of attention due to its ease of use, environmental safety, and straightforward nature. This study focuses on current developments in the biosynthesis of Metal oxide nanoparticles (MONPs) using algal extracts, which have a low environmental impact [11].

Aquatic plants (microalgae and seaweeds) are known to function as biological factories by using their dried and living biomass to produce metallic nanoparticles [12]. Both microalgae and seaweeds can grow in soil that is usually not suitable for other plants. They have the highest photosynthetic efficiency and can create the most biomass [13]. Algae MONPs are considered environmentally friendly, stable in solution, and safe for use in various fields. They have gained widespread attention and have been effectively used in areas such as sensing, catalytic chemistry, biology, and medicine. Additionally, algal cells are abundant biomaterials used in food, fertilizers, bioenergy, and wastewater treatment [14–19] due to the presence of proteins, polysaccharides, and lipids with various functional groups on the surfaces of their cell walls [13]. Seaweeds have significant metal-binding capacities, which makes them suitable for use as metal-binding sites between the adsorbent and adsorbate [20]. These green nanoparticles are commonly used as adsorbents, photocatalysts, and antimicrobials for the removal of various contaminants [21]. Transition MONPs possess unique features that make them ready for use in various applications, such as environmental remediation, sensors, lithium-ion batteries, and catalysis. As a result, they are more suitable for use in a variety of applications than other types of materials [22].

Adsorption techniques are the most commonly used methods for treating polluted liquids, involving the accumulation of metal ions on adsorbent surfaces and pores, forming a layer that includes the ions and other contaminants. Compared to conventional metal removal techniques, adsorption has several advantages, including its low cost, the ability of various types of biomass to perform differently in different metals, regenerative capability, minimal use of energy and chemicals, no sludge production, the potential for metal recovery, and competitive performance [20]. The efficiency of adsorption is affected by factors such as the amount of adsorbent used, the interaction between the adsorbate and adsorbent, temperature, adsorbent surface area, contact time, particle size, pH, and initial pollutant concentration [23].

Various industries, including those involved in mining operations, metal plating, and battery production, release toxic heavy metal effluents such as Cr, Fe, Ni, Cu, Zn, Hg, and Pb into the environment [24,25]. These metals do not biodegrade and tend to accumulate in aquatic environments, posing a significant threat to human health and the environment. When their concentration exceeds the permissible levels set by the World Health Organization (WHO) and the US Environmental Protection Agency, they are considered contaminants. Therefore, it is crucial to find effective adsorbents and techniques to remove them from water and wastewater. However, there have been few studies on the use of green ZnO-NPs produced by algae for removing inorganic pollutants through the adsorption technique [26]. There are several methods available to investigate the properties of nanoparticles at a microscopic level, including Atomic Force Microscopy (AFM), Scanning Electron Microscopy (SEM), Transmission Electron Microscopy (TEM), Field Emission Scanning Electron Microscopy (FESEM), and High-Resolution Transmission Electron Microscopy (HR-TEM) [27]. These techniques are utilized to evaluate the dimensions, morphology, and other topographic and surface characteristics of nanoparticles. These methods are vital for examining the physical and chemical attributes of NPs. They provide comprehensive details regarding the surface properties of particle size and shape, which are essential to comprehend their actions and possible applications [28].

The synthesis of MONPs involves mainly physical and chemical processes, which utilize highly hazardous and reactive reducing substances, such as sodium borohydride and hydrazine hydrate, that have negative impacts on the environment, plant life, and animal life [29]. However, the use of complex equipment, difficult processes, and harsh experimental conditions pose significant challenges [30]. Physical methods typically result in the production of heterogeneous NPs with high energy requirements, while chemical techniques use artificial capping, reducing, and stabilizing agents to generate homogeneous metallic NPs with fewer environmentally harmful byproducts [31]. MONPs are extensively used in wastewater purifications due to their large surface area and small diameters, which provide them with strong adsorption reactivity and capacity [32].

Various types of nanoparticles such as carbon nanotubes, zerovalent nanoparticles, nanocomposites, and metal oxide nanoparticles exhibit great potential for use in different applications, including wastewater ecosystems [33] and antioxidant activities [34]. In a study by Khilji et al. [35], ZnONPs derived from *Oedogonium* sp. algae were tested for their ability to eliminate heavy metals from the wastewater used in leather production. Leather wastewater contained high concentrations of Cr, Cd, and Pb before the introduction of nanoparticles. After 45 days of treatment with 1 mg of nanoparticles, the removal rates of TDS, chlorides, Cr, Cd, and Pb increased by 46.5%, 43.5%, 54%, 57.6%, and 59.3%, respectively.

The study conducted by Sunny et al. [36] investigated the use of green synthesis as a sustainable method for synthesizing tin oxide nanoparticles ($SnO_2$NPs) using a seaweed mixture of *Padina gymnospora* and *Turbinaria ornata*. The antibacterial and antioxidant properties of the synthesized $SnO_2$NPs were tested using well diffusion and DPPH scavenging methods, respectively. The $SnO_2$NPs were characterized using XRD, FTIR, UV-Vis, and SEM. The crystalline average size of the $SnO_2$NPs synthesized from *Padina gymnospora* and *Turbinaria ornata* was 3.7 nm and 24.6 nm, respectively. The synthesized NPs showed good antibacterial activity against *Escherichia coli* and *Bacillus subtilis*, with the maximum zone of inhibition observed at 75 $\mu$g mL$^{-1}$ and 100 $\mu$g mL$^{-1}$. The $SnO_2$NPs demonstrated good binding energy against PBP1a, PBP1b, PBP2d, and PBP4, indicating their potential as inhibitors for these proteins. This study suggests that $SnO_2$NPs using green synthesis can be used as a potential antibacterial agent.

Alumina ($Al_2O_3$) is one of the many phases of aluminum oxide. It is one of the numerous phases of aluminum oxide, which is one of the most widely manufactured chemicals in nanosized particles. $Al_2O_3$NPs have a large surface area, low cost, high surface reactivity, and powerful adsorption ability. The nanoparticles of aluminum are hydrophilic. Therefore, it is crucial to chemically or physically alter the surface of $Al_2O_3$ nanoparticles with specific functional groups, such as oxygen, nitrogen, sulfur, and phosphorus [37]. Koopi and Buazar [38] used the brown marine alga *Sargassum ilicifolium* to create green $Al_2O_3$NPs. The *S. ilicifolium* extract serves as a stabilizing and bioreducing agent. The absorption peak (at 227 nm) showed that the phytochemicals in the *S. ilicifolium* extract may convert aluminum ions ($Al^{3+}$) to $Al_2O_3$ nanoparticles. The $Al_2O_3$NPs were spherical, and their size ranged from 21 to 42 nm. The hydroxyl and carbonyl groups of the *S. ilicifolium* extract improved the economics of the $Al_2O_3$NPs.

The treatment of wastewater and contaminated sites is a critical aspect of environmental management. Traditional methods of treating wastewater and remediating contaminated sites are often time-consuming, costly, and require significant resources. However, recent advancements in IoT and ML techniques have shown great potential for revolutionizing the way we approach these challenges [39]. Since its inception in 1958, artificial intelligence has undergone significant development, evolving into a field that includes various technologies and machinery that have inspired other technologies [40].

The objectives of this review are to promote the use of algal extracts for the cost-effective, environmentally friendly synthesis of MONPs. The review will cover the mechanisms, characterization techniques, and properties of MONPs. It will also explore their potential applications in wastewater treatment and their benefits. Additionally, the review will discuss the adsorption and photocatalytic mechanisms of MONPs and their efficacy in organic degradation. This will help to better understand the potential of green chemistry for sustainable and effective wastewater treatment. Overall, the revised statement emphasizes the importance of using green chemistry methods for eco-friendly nanoparticle synthesis and provides a clearer overview of the review's goals and objectives, highlighting the potential application of using the Internet of Things (IoT) and Machine Learning (ML) techniques.

## 2. Green Synthesis

Green synthesis is promoted as an environmentally benign alternative to the chemical process of producing nanoparticles. The term "green synthesis" refers to methodologies, approaches, and processes that synthesize nanoparticles with the least amount of energy consumption while avoiding harmful byproducts through regulation, control, remediation, and cleanup. Waste prevention and minimization, a decrease in by-products and pollution, the use of greener (or nontoxic) solvents and auxiliaries, and renewable feedstock are the guiding principles for green synthesis [41]. The synthesis of metallic nanoparticles is better achieved using green synthesis techniques because reduction and capping agents are required during the synthesis process. Biological sources for the environmentally friendly synthesis of nanoparticles comprise bacteria, actinomycetes, fungi, yeast, algae, and plants [42]. The complete cell (of a plant, algae, or microbes) or an extract can be used as a reducing medium to produce nanoparticles and as a capping medium to stabilize these nanoparticles [43]. Depending on the principle of "green synthesis", approaches that are beneficial to the environment must be used to create chemicals and compounds. This kind of synthesis has been adopted in several instances. The three main focusses of green NP synthesis are as follows: (i) using safe solvents and reagents for synthesis; (ii) using an energy-efficient bioprocess (also known as a biosynthetic method) based on NP production; and (iii) creating NPs that are safe and environmentally friendly for use in subsequent processes. According to Hassaan et al. [44], there are numerous ways that have been used to synthesize green nanoparticles, including ultrasound, mailing, microwave, and microbes.

## 3. Characterization of Algae-Synthesized MONPs

Characterization is the description of a nanoparticle's post-synthesis properties, such as its shape, size, surface plasmon resonance, polydispersity, capping agents, crystallinity, and surface charge. Designing a nanomaterial with the necessary practical applications depends on all these physical and chemical properties [5,6,45,46]. Figure 1 displays the various methods of MONPs characterization for the investigated properties.

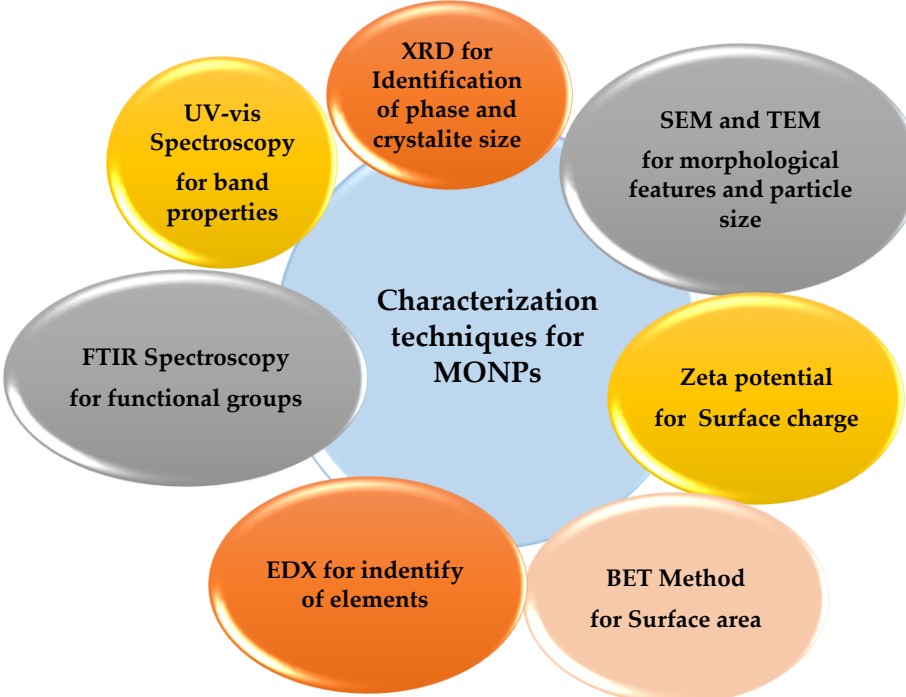

**Figure 1.** Simplified graph of the general characterization techniques of MONPs.

## 4. Factors Affecting the Synthesis of Green NPs

A few of the factors that affect the production of nanoparticles using algal extract include pH, temperature, exposure time, reaction time, and stirring speed. Another factor is the quantity of bioactive compounds in the algal extract. All of these factors help to determine how quickly algal nanoparticles are produced as well as their size, shape, characteristics, and functions [47].

### 4.1. pH

The pH value must be set to prevent the preparation of NPs with various shapes and sizes throughout the synthesis process. Changes in the pH of the reaction may have an impact on the shape and size of the produced nanoparticles. Larger particles are produced when comparing lower acidic pH values to higher acidic pH values [11]. In contrast, a high pH environment is ideal for the synthesis of small-sized NPs because it promotes the development of spherical NPs. A wide variety of NPs is produced when the pH is low because the surface plasmon resonance (SPR) peak expands and diffracts toward a longer wavelength area (mostly cylindrical or triangular-shaped) [48].

### 4.2. Temperature

Temperature is an important key that controls how NPs are created. The creation of nanoparticles utilizing biological components has shown catalytic behavior as the temperature has increased the reaction rate and effectiveness. The reaction temperature has a significant impact on several chemical processes (e.g., electrochemical, solvothermal, or templating methods) [49]. The temperature needed to synthesize NPs using green techniques is often close to 100 °C. Temperatures more than 350 °C are required for physical reactions, although only moderate temperatures are required for chemical synthesis. However, as their conversion rate increased, their average size decreased. The researchers hypothesized that high temperatures may have had an impact on an essential enzyme engaged in the production of nanoparticles [11].

### 4.3. Concentration of Extract

Algal extract constituent performs the dual functions of bioreductant and stabilizer of produced nanoparticles. Nanoparticle properties are influenced by the quantity of bioreducing agents in an algal extract by controlling the size and shape of the particle. As the concentration of the reducing agent in the algal extract increased, the nucleation rate increased. However, high synthesis rates at high extract concentrations result in smaller nanoparticles [50]. According to Aboelfetoh et al. [51], increasing the extracted content of *Caulerpa serrulata* from 10% to 20% resulted in a reduction in the size of silver nanoparticles from 21–30 nm to 8–10 nm. Furthermore, extract concentration changes the morphology of all ellipsoids, nanocubes, and nanorods to spheres at 10% extract concentration, unlike conventional spherical-shaped ZnNPs at a lower concentration of *S. myriocystum* extract. The study by Nagarajan et al. [52] reported that increased extract concentrations led to the generation of zinc NPs with irregular shapes such as hexagonal, triangular, rod, and radial particles. Because a higher extract concentration may lead to nanoparticle aggregation, an optimal algal extract concentration is necessary for the efficient synthesis of nanoparticles.

### 4.4. Contact and Reaction Times

Reaction time is the amount of time needed for a reaction to finish or for a nanoparticle to develop. It typically gauges how quickly nanoparticle synthesis takes place. The duration of the reaction, or the amount of time that metal salts are exposed to algal extract, is measured as the "contact time". Algal extract concentration has a significant impact on both reaction and contact time. Algal extracts with higher concentrations of the reductive group tend to produce nanoparticles more quickly. *Padina pavonia* algal extract was used by Abdel-Raouf et al. [53] to synthesize silver nanoparticles in just two minutes, as opposed to three hours for the formation of *P. pavonia* powder.

*4.5. Stirring Rapidly*

The rate of stirring has a considerable impact on particle size. We see the smallest nanoparticles with the fastest stirring speed (1000 rpm) and the largest nanoparticles with the slowest (100 rpm). Higher stirring rates are associated with greater energy dissipation in the system, which is associated with faster seeding and hence more green materials when compared to slower stirring rates [54]. Forge et al. [55] noticed the same phenomenon and proposed the largest iron oxide nanoparticles by employing a slow stirring speed and a high temperature. According to Deni Subara [56], the size decreased as the stirring speed climbed from 150 rpm to 600 rpm, which is the highest permissible stirring speed in our experimental setup. However, there was no discernible influence of stirring speed on nanoparticle size. The increase in stirring speed causes an increase in shearing energy that breaks the large particles into smaller particles, which could prevent massive agglomeration, but further increasing the stirring rate has no effect on particle size [56]. As a result, the stirring speed should be kept around 600 to produce smaller nanoparticle sizes. In addition, the study by Bhattacharya et al. [57] noted that at 900 rpm, the copper oxide nanoparticles shrink in size from 42.54 nm to 3.6 nm (at 500 rpm). Furthermore, the smaller size may be attributed to the bigger nanoparticles being broken up into smaller ones at a faster stirring speed. Increased nanoparticle synthesis rate brought on by uniform distribution of metal salts and bioreductant species of the algal extract is another benefit of faster stirring. Homogeneous distribution speeds up the reduction of metal salts to nanoparticles by enhancing the interaction between species of bioreductant and metal salts.

*4.6. Choice of the Organism or Species*

To achieve a cost-effective synthesis of nanoparticles, it is crucial to consider the physical–chemical parameters mentioned earlier, as well as other factors such as (i) selecting the appropriate organism or species based on their intrinsic characteristics, such as growth rate, metabolic pathways, and enzyme activities; (ii) the inoculum size; and (iii) the choice of biocatalysts to accelerate the reaction rate (i.e., reduction). However, the latter should be considered in the presence of other factors. Biocatalysts such as enzymes, raw cells, or whole cells (e.g., NADPH, FAD) can be used. Whole cells are preferred due to their cost-effectiveness, as coenzymes can be recycled along the pathway, making them highly efficient [58].

## 5. Green Synthesis of MONPs from Algae

Seaweed is a common name for macroalgae, which are classified based on their cell wall chemistry, flagella presence, and other characteristics. The most prominent feature utilized for categorizing macroalgae is the presence of pigments, apart from chlorophyll, which distinguishes them as green algae, brown algae, or red algae—the primary types of algae. These classifications may have biological activity. Microalgae are the simplest and most primitive members of the plant kingdom, with cell sizes ranging from 3 to 20 m. They are present in coastal and bottom habitats and as phytoplankton throughout oceans and aquatic ecosystems [15,19].

*5.1. The Main Chemical Components of Marine Algae*

Several chemical components that can be extracted from macro- and microalgae include polysaccharides, vitamins, lipids, proteins, dietary fiber, antioxidants, minerals, and polyunsaturated fatty acids [59]. To give a brief example, among the potential chemical compounds effective in cancer treatments are some phytochemicals, such as carotenoids, pigment, scytonemin, several calothrixins, and phlorotannins [60]. Brown algae can also be used to extract other polysaccharides, such as alginates and fucoidans. Brown algae also produced Laminarins, which are short-chain polysaccharides with a 1, 3-D-glucopyranose backbone and a minimal amount of side branching (−1, 6-D-glucopyranose units). Several authors have also highlighted the use and advantages of alginate-based nanocomposite materials for applications such as the production of biomaterials and the removal of

aqueous dyes [61]. Table 1 illustrates the most numerous compounds adapted for the algae synthesis of NPs.

**Table 1.** The main bioactive substances extracted from algal sources involved in NPs synthesis.

| Phytochemicals Identified | Algae Phytochemicals | Refs |
|---|---|---|
| Proteins | Differences in its lower and higher levels | [62] |
| Polysaccharides | Galactans | [63] |
| | Fucoidan | |
| | Laminarin | |
| | Alginates | |
| | Carrageenan | [64] |
| | Agar | |
| Pigments | Chlorophylls | [65] |
| | Carotenoids | |
| | Phycobiliproteins | |
| Polyphenols | Lignans | [66] |
| | Flavonoids | |
| | Quercetin | |
| | Cinnamic acid | |
| | Benzoic acid | |
| | Isoflavones | |
| | Phenolic acids | |
| Vitamins | Retinol | [67] |
| | Niacin | |
| | Riboflavin | |
| | Thiamine | |
| | Pantothenic acid | |
| | Pyridoxine | |
| | Biotin | |
| Lipids | Glycolipids | [68] |
| | Phospholipids | |
| Fatty acids | Polyunsaturated fatty acids | [69] |
| | Monounsaturated fatty acids | |

*5.2. Methodology for Synthesizing Algae-MONPs*

The utilization of algae in the development of safe, biocompatible, non-toxic, cost-effective, and environmentally friendly synthesis methods is on the rise. With many algae species that have not yet been explored for biogenic synthesis, there is still potential for further research in this area [70]. Metallic nanoparticles are created using "top-down" and "bottom-up" methods [22]. In order to produce nanosized particles, the top-down approach uses various techniques for the size reduction of bulk material. In the "bottom-up" method, atoms self-assemble on top of one another to form crystal planes; these planes then stack on top of one another to form nanostructures or nanoparticles [71] The "bottom-up approach" describes how metal salts are synthesized into metal nanoparticles in the presence of biological cells, extracts, or their components in the green synthesis process of producing metal and MONPs. The bottom-up method is the preferred approach (whether acidic,

basic, or neutral) due to its superior efficacy compared to process variables such as solvent, temperature, pressure, and pH conditions [72].

*5.3. Algal Extract Preparation*

The algal extract is used to create nanoparticles in phyconanotechnology, which is a more recent field of nanoscience. Algal extracts are used because they are less poisonous, easy to handle, and can develop at low temperatures. Algal phytochemicals function as an efficient metal-reducing and capping agent to create a durable coating on the metal NPs in one-step production. Algae are a common and valuable source for the synthesis of metallic NPs due to their abundance and ease of accessibility as materials [70]. The study by Yew et al. [73] highlighted the potential of algae as a sustainable source for the production of metal oxide nanoparticles, which have various applications in fields such as medicine, electronics, and environmental remediation. By using algae instead of traditional chemical synthesis methods, this process offers a more environmentally friendly approach to producing nanoparticles. Using the findings of Yew et al. [73], the production of metal oxide nanoparticles by algae can be successively achieved. To elaborate further on the process, the algae used in the study were first collected and carefully cleaned to ensure that any external impurities were removed. Once clean, the algae were ground into a fine powder, which was then steeped in deionized water for several hours to allow for the extraction of active compounds. The resulting extract was then filtered using filter paper to remove any remaining impurities, resulting in a pure solution. To produce metal oxide nanoparticles, a salt solution containing metal ions was added to the purified algae extract. The metal ions reacted with the active compounds extracted from the algae, resulting in the formation of metal oxide nanoparticles [73].

*5.4. Mechanism of Nanoparticle Synthesis by Algae or Algal Extract*

The study by Fawcett et al. [72] reported that there are two methods for the synthesis of metallic nanoparticles: intracellular and extracellular. The extracellular technique employing algal extract is the method of choice for synthesis because of its simplicity to handle, cost-effectiveness, and lack of the additional step of disrupting cells to recover internal nanoparticles for downstream processing. An extracellular technique of synthesis is the conversion of metal salts to metal nanoparticles using a suspension of algal extract. Algal extracts are a rich source of many bioactive compounds; these chemical compounds act as reducing and capping agents when nanoparticles are produced [74]. Metal nanoparticles are created by the reduction of metal salts through bioactive substances released by algae cells in extract media. The two stages of nucleation and crystal growth are said to be involved in the production of nanoparticles. The precursor metal salts ($M^+$) are converted to metal nanoparticles ($M^0$) during the nucleation reaction by an ingredient in an algal extract. Crystal growth is defined by the complexation of metal nanoparticles ($M^0$) with other metal salts ($M^+$) and the formation of the $M^0$-$M^+$ complex in the second stage, which is followed by adsorption for the generation of nanocrystals as metal nanoparticle dimers. In intracellular synthesis, metal salts are reduced inside of the cells with the help of nitrate reductase; NADH- and NADPH-dependent enzymes; enzymes involved in metabolic processes including photosynthesis, nitrogen fixation, and respiration; and algal pigments such as chlorophyll [75]. Both the living and dry biomass of algae are capable of performing the intracellular production of nanoparticles. Intracellular synthesis is the result of an algal cell's defensive mechanism against a stress condition produced by metals, which diverts the algal cells' metabolic machinery to reduce the toxic effects of metals. The process of creating nanoparticles inside cells involves binding positively charged metal ions to the cell wall's surface or the cytoplasm's negatively charged protein or enzyme groups [76].

*5.5. Synthesis of Various MONPs by Marine Algae*

The biotechnological developments that explain the producing of MONPs using algae have been the focus of this section of the review. Table 2 provides a brief summary of the

numerous studies encountered regarding the synthesis of metal oxide nanoparticles by marine algae.

**Table 2.** Synthesis of MONPs using several algal species.

| Algae Strain | MONPs | Size (nm) & Shape | Ref |
|---|---|---|---|
| *Sargassum muticum* | ZnO | 30–57 nm | [77] |
| *Spirulina platensis* | $TiO_2$ | 300 nm | [78] |
| *Bifurcaria bifurcata* | $Cu_2O$ | 5–45, spherical | [79] |
| *Bifurcaria bifurcata* | CuO | 96–110, spherical | [80] |
| *Kappaphycus alvarezii* | $Cu/Cu_2O$ | 53, spherical | [81] |
| *Anabaena cylindrical* | CuO | 3.6/rod | [57] |
| *Cystoseira trinodis* | CuO | 6–7.8 nm | [82] |
| *Botryococcus braunii* | CuO | 7–10 nm | [83] |
| *Sargassum polycystum* | CuO | 17 mm | [84] |
| Red seaweed | $Co_3O_4$ | >30 | [85] |
| *Sargassum muticum* | $Fe_3O_4$ | 18 ± 4, cubic | [86] |
| *C. sinuosa, P. capillacea* | $Fe_3O_4$ | 11.2–33.7 nm | [87] |
| Brown seaweed | $Fe_3O_4$ | 10–19.5 nm | [88] |
| *Chlorella vulgaris* | $CuFe_2O_4@Ag$ | 20 nm | [89] |
| *Spirulina platensis* | $Fe_3O_4@Ag$ | 30–68 nm | [90] |
| *Spirulina platensis* | $Fe_3O_4/Ag$ | 30–50 nm | [91] |
| *Spirulina platensis* | $Fe_3O_4$ | | [91] |
| *Cystoseira myrica* | CuO | 11–80 nm | [92] |
| *Kappaphycus alvarezii* | $Fe_3O_4$ | 14.7 nm Spherical | [73] |
| *Chaetomorphaantennina* | $Fe_3O_4$ | 9–10 nm | [93] |
| *Turbinriatur binata* | $Fe_3O_4$ | 8–14 nm | [93] |
| *Pterocladia capillacea* | $Fe_3O_4$ | 16.85–22.47 nm Nanosphere | [94] |
| *Jania rubens* | $Fe_3O_4$ | 22.22–33.33 nm, Spherical | [95] |
| *Colpomenia sinuosa* | $Fe_3O_4$ | 11.24–33.71 nm Nanosphere | [87] |
| *Dictyota indica* | PdO | 8–43 nm Spherical | [96] |
| *Sargassum wighitii* | MgO | 68.06 nm | [97] |
| *Spirogyra* sp. | NiO | 27.7 nm | [98] |
| *Spirulina platensis* | $Fe_3O_4$ | – | [91] |
| *Cystoseira crinita* | MgO | spherical with sizes of 3–18 nm | [37] |
| *Macrocystis pyrifera* | CuO | Between 2 and 50 nm | [99] |

### 5.5.1. Copper Oxide Nanoparticles (CuONPs)

Copper oxide nanoparticles (CuONPs), sometimes referred to as cupric oxide or copper (II) oxide, are a monoclinic semiconducting chemical. Because CuO is the most basic copper compound and has several interesting physical properties, including high temperature, superconductivity, electron correlation effects, and spin dynamics, it has been the subject of a significant amount of studies. CuONPs have applications in solar energy and conversion systems, batteries, high-temperature superconductors, gas sensors, and antimicrobials in addition to water purifiers, algaecides, fungicides, antibacterial, and anti-fouling agents, among other uses. *Anabaena cylindrica* extract was utilized by Bhattacharya et al. [57] to create rod-shaped copper oxide nanoparticles (CuONPs), which were employed to disinfect

drinking water. Cu$_2$ONPa production was suggested by the development of a murky, brownish solution. In another study, the algal extract of *Brassica oleracea* can be used to create CuONPs, as reported by Renuga et al. [100]. Functional groups from polyols, such as flavones and terpenoids, and polysaccharides, such as hydroxyl, carbonyl, and amine, were thought to be responsible for the reduction, capping, and stability of the CuONPs produced. In another study, Araya-Castro et al. [99] synthesized CuONPs using protein fractions from the water extract of the brown alga *Macrocystis pyrifera*. With the use of a UV/VIS diode array, proteins were identified. Time-based fraction collection was then completed, and each fraction was used to assess the production of CuONPs. A TEM fitted with an energy-dispersive EDS detector was used to assess the characterization of CuONPs. Both the low-molecular-weight and high-molecular-weight protein fractions were capable of producing spherical CuONPs. The brown algae (*Bifurcaria bifurcata*) extract was used by Abboud et al. [80] to biosynthesize CuONPs. Both Cu$_2$O and CuO were synthesized using the same simple technique. Aqueous copper (II) sulfate solution was reduced to CuONPs by the diterpenoids in brown algae extract belonging to the unsaturated ketone group. A minor percentage of the nanoparticles were slightly elongated, but the bulk was spherical. The mean particle size of the nanoparticles, which ranged in size from 5 to 45 nm, was found to be 22.6 nm. Red seaweed (*Kappaphycus alvarezii*) extracts were used in a recent study by Khanehzaei et al. [81] to biosynthesize CuONPs. The CuONPs were spherical and had a mean particle size of 53 nm when seaweed was used as a stabilizing agent. Their research revealed that the water-soluble sulfated polysaccharides generated from the seaweed's cell walls, together with certain hydroxyl and sulfate groups, were used to cap the nanoparticle surface.

### 5.5.2. Zinc Oxide Nanoparticles (ZnONPs)

ZnONPs are n-type semiconducting MOs with several applications in biomedical systems, cosmetics, rubber, electronics, and rubber products. Priyadharshini et al. [101] utilized an extract of *Gracilaria edulis* to generate ZnONPs. The extract of *G. edulis* was capable of converting the aqueous Zn$^{2+}$ solution to ZnONPs, resulting in a color change from white to reddish brown in the precursor solution. The rod-shaped nanoparticles, which were coated with quinine, exhibited enhanced cellular toxicity against the PC$_3$ human malignant cell line due to their ability to penetrate cells more effectively and cause cell death, thanks to their larger surface area for adhesion to cells. Another approach to synthesizing ZnONPs was described by Francavilla et al. [102], who utilized a reactive milling process with agar derived from *G. gracilis* as a template material. During milling, the zinc precursor Zn(NO$_3$)$_2$ was transformed into a highly crystalline hexagonal wurtzite structure, and the resulting nanoparticles had sizes ranging from 18 to 50 nm. Following calcination at 600 °C, the porous ZnONPs demonstrated excellent photocatalytic capabilities, making them suitable for the breakdown of aqueous phenol solutions. Shariati et al. [103] studied the use of functionalized indium-doped zinc oxide nanoparticles (In-ZnONPs) to inhibit microalgal (*Chlorella vulgaris* and *Scenedesmus quadricauda*) growth in order to preserve adobe mud and earthen-made artworks in humid conditions. The NPs were synthesized and analyzed using SEM, FESEM, and HRTEM, and their effect on algal growth was tested using a growth inhibition test. The results found that at a concentration of 200 mg L$^{-1}$, ZnONPs had a growth inhibition rate of up to 100%, damaging the structure and function of algal cells. The study suggests that In-ZnONPs have potential applications in preserving adobe mud and earthen-made artworks in humid conditions. Vasistha and Rai [104] utilized microalgae *Chlorosarcinopsis* sp. to synthesize ZnO-NPs and treat wastewater for the production of biodiesel. The microalgae were grown in primary-treated wastewater and combined with ZnO-NPs in secondary-treated wastewater to achieve high removal efficiencies of total nitrogen (87.20%), total organic carbon (97.5%), and total phosphorus (82.21%) at a concentration of 10 mg L$^{-1}$ ZnO-NPs. Another study conducted by the same authors employed ZnO-NPs synthesized using Oedogonium sp. algae to remove heavy

metals from leather effluents. After 45 days of treatment with 1 mg of nanoparticles, TDS, chlorides, Cr, Cd, and Pb removal efficiencies were increased.

### 5.5.3. Iron Oxide Nanoparticles (FeONPs)

Recently, the biosynthesis of iron oxide nanoparticles ($Fe_3O_4NPs$) using seaweeds was reported, as shown in Table 3. This method is simple, pollution-free, friendly to the environment, and inexpensive [87]. The photosynthesis of $Fe_3O_4NPs$ was achieved by using the algal extract as a reductant of $FeCl_3$. The phytogenic $Fe_3O_4NPs$ were characterized by a surface plasmon band observed close at 402 nm and 415 nm. The obtained $Fe_3O_4NPs$ had particle sizes of 10–19.5 nm for *P. pavonica* and 21.6–27.4 nm for *S. acinarium*. According to their respective EDX spectra, the iron signals were reported to be strong. Sulfated polysaccharides were shown to be the primary macromolecules in the algae extracts, with the dual ability to reduce $FeCl_3$ and stabilize phytogenic $Fe_3O_4NPs$, according to FTIR studies. Mahdavi et al. [86] found that brown seaweed could be used to create ferric oxide $Fe_3O_4NPs$ in a single step using green biotechnology (*Sargassum muticum*). To create $Fe_3O_4NPs$, the aquatic seaweed extract was combined with an aquatic ferric chloride solution. It has been found that the amino, carboxyl, and hydroxyl functional groups of cell walls' water-soluble polysaccharides can function as both a capping and a reducing agent. The particles were cubic and crystalline, and their average size was 18.4 nm. The study by El-Sheekh et al. [105] synthesized $Fe_3O_4NPs$ using two brown seaweed species of *Padina pavonica* and *Sargassum acinarium* and water extracts as a green biosynthetic method. The obtained $Fe_3O_4NPs$ were characterized by their particle size and iron content. Sulfated polysaccharides from seaweed extracts were found to have a dual role in reducing $FeCl_3$ and stabilizing phytogenic $Fe_3O_4NPs$. The synthesized $Fe_3O_4NPs$ were then utilized in Pb adsorption tests. Interestingly, the $Fe_3O_4NPs$ alginate beads, which were synthesized via *P. pavonica*, exhibited a remarkable lead bioremoval capacity of 91% in just 75 min. The study by Hassan et al. [106] evaluated the efficacy of FeONPs synthesized through green methods in the removal of pyrene and benzo(a)pyrene (PAHs) micropollutants from water. The study examined the impact of several factors, such as the dose of nanoparticles, pH, temperature, and initial PAH concentration, on the adsorption process. The researchers found that the maximum adsorption capacity of green iron oxide nanoparticles for pyrene and benzo(a)pyrene was 2.8 and 0.029 mg/g, respectively. In another study, green iron nanoparticles with a spherical shape and a size range of 5–10 nm were produced for hydrocarbon treatment. The researchers investigated the ability of nanoparticles to remove total petroleum hydrocarbons (TPH) from contaminated soil and water and found that the iron nanoparticles removed 88.2% of TPH from water samples after 12 and 32 h of treatment. El-Sheekh et al. [107] studied the utilization of biosynthesized iron oxide nanoparticles from aqueous extracts of three brown seaweed species, *Padina pavonica*, *Colpomenia sinuosa*, and *Petalonia fascia*, to address marine pollution caused by excess nutrients, such as nitrogen and phosphorus, and harmful algae. The study used aqueous extracts from three species of brown seaweed to biosynthesize $Fe_3O_4NPs$. The results showed that the $Fe_3O_4NPs$ synthesized via *P. pavonica* had the highest percentage of nitrogen reduction, while *P. fascia* and *P. pavonica* were effective at removing phosphorus and reducing Chl-a. The study suggests that biosynthesized iron oxide nanoparticles using brown seaweed extracts could be a promising approach to mitigate marine pollution.

**Table 3.** Metal oxide-based nanomaterials for the removal of different pollutants in wastewater by adsorption technique.

| Algal Species | Adsorbents (MONPs) | Adsorbate/ Pollutants | Optimized Conditions | Method/Efficiency, Adsorption Capacity $(mg\ g^{-1})$, Removal (%) | Refs |
|---|---|---|---|---|---|
| *Spirulina platensis* | $Fe_2O_3NPs$ | Crystal violet dye | $256.4\ mg\ g^{-1}$ | 180 min; 25 °C; pH 5 | [108] |
| *Spirulina platensis* | $Fe_2O_3NPs$ | Methyl orange | $270.2\ mg\ g^{-1}$ | 180 min; 25 °C; pH 5 | [108] |
| *Pterocladia Capillacea* | ZnONPs | Ismate Violet 2R | 59.88% | 60–120 min; pH 2 | [7] |
| *Spirulina platensis* | $Fe_2O_3NPs$ | Crystal Violet | 55.62 mg/g | 35 °C; pH 12 | [91] |
| *Dictyota indica* | PdONPs | Cadmium | 82.82% | pH 8; 20 min; $9\ g\ L^{-1}$ | [96] |
| *Padina pavonica* | $Fe_3O_4NPs$ | Lead | (91%) | pH 6 | [88] |
| *Sargassum acinarium* | $Fe_3O_4NPs$ | Lead | (78%) | pH 6 | [88] |
| *Synechocystis* sp. | $Fe_2O_3NPs$ | Cr(VI) | pH 2 | $69.77\ mg\ g^{-1}$ | [109] |
| *Synechocystis* sp. | $Fe_2O_3NPs$ | Cu(II) | pH 5.3 | $38.68\ mg\ g^{-1}$ | [109] |
| *Synechocystis* sp. | $Fe_2O_3NPs$ | Pb(II) | pH 5.3 | $62.63\ mg\ g^{-1}$ | [109] |
| *Synechocystis* sp. | $Fe_2O_3NPs$ | Cd(II) | pH 5.3 | $42.12\ mg\ g^{-1}$ | [109] |
| *Petalonia fascia* | $Fe_3O_4NPs$ | Total Nitrogen | 10 days | 72.74% | [107] |
| *Petalonia fascia* | $Fe_3O_4NPs$ | Total Phosphorous | 10 days | 71.5% | [107] |
| *Petalonia fascia* | $Fe_3O_4NPs$ | Chlorophyll a | 10 days | 94.4% | [107] |
| *Colpomenia sinuosa* | $Fe_3O_4NPs$ | Total Nitrogen | 10 days | 83.97% | [107] |
| *Colpomenia sinuosa* | $Fe_3O_4NPs$ | Total Phosphorous | 10 days | 67.62% | [107] |
| *Colpomenia sinuosa* | $Fe_3O_4NPs$ | Chlorophyll a | 10 days | 94% | [107] |
| *Padina pavonica* | $Fe_3O_4NPs$ | Total Nitrogen | 10 days | 89.8% | [107] |
| *Padina pavonica* | $Fe_3O_4NPs$ | Total Phosphorous | 10 days | 93.96% | [107] |
| *Padina pavonica* | $Fe_3O_4NPs$ | chlorophyll a | 10 days | 89% | [107] |
| *Padina pavonica* | $Fe_3O_4NPs$ | $NH_4$ | 13 h; pH 7.0 | 85% | [107] |
| *Anabaena cylindrica* | CuONPs | Detoxification | – | – | [57] |
| *Calotropis procera* | CuONPs | Cd(II) | pH 6; 20 min; 0.4 g | 84.75 | [110] |
| *Calotropis procera* | CuONPs | Fe(III) | pH 6; 20 min; 0.4g | $126.32\ mg\ g^{-1}$ | [110] |
| *Chlorosarcinopsis* sp. | *ZnONPs* | Organic carbon | $10\ mg\ L^{-1}$ | 97.5% | [104] |
| *Chlorosarcinopsis* sp. | *ZnONPs* | Total nitrogen | $10\ mg\ L^{-1}$ | 87.20% | [104] |
| *Chlorosarcinopsis* sp. | *ZnONPs* | Total phosphorous | $10\ mg\ L^{-1}$ | 82.21% | [104] |

### 5.5.4. Magnesium Oxide Nanoparticles (MgONPs)

Magnesium is an essential mineral for the growth of plants. It serves as a driving force in the photosynthetic process. Furthermore, it shows that the production of nanoparticles involves a robust interaction with phytoconstituents found in algae and plants [111]. Magnesium oxide nanoparticles (MgONPs), which have great antibacterial capabilities, are a very safe and acceptable alternative, according to the FDA. MgONPs may have a long-lasting antibacterial effect due to their low volatility and excellent temperature stability [112]. Pugazhendhi et al. [97] employed *S. wighitii* as a capping and reducing agent to synthesize MgONPs. The MgONPs were characterized in their as-prepared state using microscopic investigations, revealing a strong absorption peak at 322 nm in their UV-visible spectra. X-ray diffraction analysis confirmed the crystalline nature of the MgONPs with a face-centered cubic structure, and the presence of magnesium and oxygen was verified by EDX profiling. The FTIR data suggested the involvement of unique functional groups

present in sulfated polysaccharides in the production of MgONPs. Dynamic light scattering and zeta potential analyses demonstrated that the MgONPs were highly stable, with an average size of 68.06 nm and a zeta potential of 19.8 mV. The MgONPs exhibited potent antibacterial, antifungal, and photocatalytic activity and cytotoxicity against A549 (lung cancer cell lines) in a dose-dependent manner, with an $IC_{50}$ value of $37.5 \pm 0.34$ µg mL$^{-1}$. Safety evaluation using PBMCs (peripheral blood mononuclear cells) illustrated the MgONPs to be non-toxic. The study suggests that marine algae-based MgONPs have the potential for various biological applications.

A study by Arunprasad et al. [113] investigated the potential benefits of incorporating MgONPs into *C. vulgaris* algae biodiesel to improve its performance and reduce emissions in a diesel engine. They used transesterification to produce methyl ester from the biodiesel and analyzed the morphology of the MgONPs using various techniques. The fuel properties of the biodiesel met ASTM standards, and the engine was tested under different load conditions using blends of biodiesel with and without MgONPs. The results showed that the addition of 100 ppm of MgONPs to a blend of B20 improved brake thermal efficiency and reduced specific fuel consumption compared to B20 alone. Furthermore, introducing MgONPs to the blend led to lower emissions of HC, CO, and smoke. This study highlights the potential of using MgONPs to enhance the performance and environmental impact of biodiesel fuels.

In the study by Fouda et al. [37], *Cystoseira crinite's* secreted metabolites were used as a biocatalyst to produce MgONPs through green synthesis. The resulting MgONPs were characterized using several methods, including TEM, XRD, UV, FT-IR, SEM-EDX, and XPS, which revealed that the MgONPs were spherical and crystallographic with sizes of 3–18 nm and a maximum surface plasmon resonance of 320 nm. The EDX data confirmed the presence of Mg and O in the sample, with weights of 54.1% and 20.6%, respectively. The synthesized MgONPs showed promising antimicrobial activity against several microorganisms and were found to be effective against Caco-2 (cancer cell lines) with an $IC_{50}$ value of 113.4 µg mL$^{-1}$. The produced MgONPs showed significant larvicidal and suicidal activity against Musca domestica, with $LC_{50}$ and $LC_{90}$ values ranging between 3.08 and 10.43 µg mL$^{-1}$, respectively. Additionally, MgONPs exhibited repellence activity for adults of *M. domestica* after 12–72 h.

### 5.5.5. Titanium Dioxide Nanoparticles (TiO$_2$NPs)

Titanium dioxide (TiO$_2$) is an inert, nontoxic, and inexpensive material with a high refractive index and UV absorption capacity that serves as an appealing white pigment and a sustainable and friendly catalyst [114]. Because of its high physical stability and lack of toxicity, TiO$_2$ nanoparticles are commonly used to impart whiteness and opacity in toothpaste, sunscreen lotions, paints, plastics, papers, inks, and food colorants [115]. Green TiO$_2$NPs were produced by *Sargassum myriocystum*, a macroalga, according to Balaraman et al. [116]. A previous UV-vis spectroscopy analysis supported the production of TiO$_2$NPs. The TiO$_2$NPs' crystalline structure, functional groups (phycomolecules), particle morphology (cubic, square, and spherical), size (50–90 nm), and surface charge (negative) were examined and validated by several characterization analyses. In addition, green titanium dioxide nanoparticles were synthesized by Vasanth et al. [78] using *Spirulina platensis* algae extract. The results showed that the absorbance spectra of the produced NPs were seen at 300 nm. The production of excellent crystalline TiO$_2$NPs with an anatase phase was discovered using X-ray diffraction research. The creation NPs were dispersed irregularly and had a spherical form. The green creation of TiO$_2$NPs is an inexpensive, simple, and environmentally friendly technology with potential applications in many different fields.

In the study by Hartmann et al. [117], the impact of different sizes of TiO$_2$ particles (10, 30, and 300 nm) on the freshwater species *Pseudokirchneriella subcapitata* was investigated. Although algal growth was inhibited by all three particle types, the exact mechanism of action remains unknown. Concentration dose–response relationships were established, but nanoparticle aggregation, sedimentation, and attachment to vessel surfaces affected repro-

ducibility. The presence of TiO$_2$ reduced the toxicity of cadmium to algae by decreasing the bioavailability of cadmium through sorption/complexation with TiO$_2$. However, the 30 nm TiO$_2$NPs caused greater growth inhibition than could be explained by the concentration of dissolved Cd(II) species, indicating a possible combined toxic effect of TiO$_2$ and Cd(II). These findings demonstrate the importance of systematic studies of nanoecotoxicological effects and the need to consider potential interactions with environmental contaminants when assessing the risks of nanoparticles.

### 5.5.6. Palladium Oxide Nanoparticles (PdONPs)

Potential applications for palladium oxide nanoparticles (PdONPs) include their ability to act as catalysts for a variety of organic and inorganic reactions at room temperature. Furthermore, nanoparticles are used to remove environmental toxins because the particular surface area has high reactivity and adsorption [118]. Green PdNPs were produced by Yazdani et al. [96] using *Dictyota indica* seaweed extract. There is a large order of brown algae called *Dictyotales* (class Phaeophyceae). The reaction mixture's hue altered, indicating the emergence of palladium oxide nanoparticles. The features of the produced nanoparticles were identified using SEM, UV, XRD, TEM, and FTIR. PdONPs were detected by UV-visible analysis. According to DLS, SEM, and TEM examinations, the PdONPs had a spherical shape, with an average particle diameter of 19 nm.

The study conducted by Arsiya et al. [119] examined the application of *Chlorella vulgaris* aqueous extract to the green synthesis of PdONPs. The researchers observed the synthesis of the nanoparticles in just 10 min and confirmed their presence using various techniques, including UV, TEM, FTIR, and SEM. The production of PdONPs was supported by an absorption peak detected between 410 and 420 nm using a UV-visible spectrophotometer. The TEM figures showed an average particle size of 15 nm, with some particles ranging between 5 and 20 nm. The NPs were determined to be crystalline through XRD pattern analysis. FTIR revealed the involvement of polyol and amide groups present in *C. vulgaris* in the synthesis of PdNPs. Thus, the functional groups played a crucial role in reducing the metal ions in an environmentally friendly and harmless process.

Lengke et al. [120] conducted a study to explore the interaction between *Plectonema boryanum* UTEX 485 cyanobacterial biomass and aqueous palladium(II) chloride (PdCl2°) at temperatures ranging from 25 to 100 °C for a maximum of 28 days. The study found that the release of organic matter from the cyanobacteria caused the precipitation of Pd(0) as crystalline nanoparticles with spherical and elongated shapes ($\leq$30 nm), both in solution and dispersed and attached to the cyanobacterial cells. On the contrary, when conditions were abiotic at 100 °C, the predominant form of palladium precipitated was palladium hydride (PdHx), with only a small amount of palladium metal formed.

The study by Prasad et al. [121] synthesized PdNPs using *Sargassum wightii* aqueous extract. Characterization by various techniques such as UV, TEM, FTIR, and XRD was performed. The results showed that the seaweed extract was capable of reducing palladium ions and forming nanoparticles. The nanoparticles synthesized using seaweed extract were found to be spherical, with an average size of 10–30 nm. UV-Vis spectroscopy analysis showed an absorption peak at 270 nm, which confirmed the presence of palladium nanoparticles. The study also investigated the impact of various conditions such as pH, temperature, and concentration on the synthesis of PdONPs. The optimum parameters for the PdONPs synthesis were found to be at a temperature of 60 °C under pH 9. In conclusion, the study demonstrated that the green synthesis of PdONPs using seaweed extract is environmentally friendly and cost-effective. The synthesized nanoparticles were found to be of good quality and could have potential applications in several fields, such as catalysis, sensors, and biomedical applications.

Momeni and Nabipour [122] conducted a study where they synthesized PdONPs using an extract from *Sargassum bovinum*. The compounds present in the extract were responsible for reducing palladium ions to form PdONPs. The properties of the PdONPs were confirmed using various techniques, including EDX, TEM, XRD, and FTIR. TEM analysis revealed that the PdONPs were monodispersed and octahedral in shape, with a size range of 5 to 10 nm. The researchers also tested the catalytic performance of the PdNPs by using them to electrochemically reduce hydrogen peroxide ($H_2O_2$). They developed a PdNP-modified carbon ionic liquid electrode (PdNPs/CILE) for the detection of $H_2O_2$.

## 6. Application of Algae-Based MONPs for Bioremediation

Adsorption is preferred over conventional metal removal techniques because of a number of its advantages: (a) economical: adsorbents are inexpensive, (b) metal selection: the metal-sorbing performance of various kinds of biomass can be more or less on various metals, (c) regenerative: adsorbent can be reused after the metal has been recycled, (d) uses less energy as well as chemical consumption, (e) no production of sludge, (f) there may be metal recovery, and (g) competitive performance: it can performance comparably. Factors that influence the adsorption efficiency comprise the amount of adsorbent, adsorbate–adsorbent interaction, temperature, adsorbent surface area, contact time, particle size, pH, initial pollutant concentration, etc. [123]. Nanoparticles used as adsorbents should be non-toxic, have a high adsorption capacity, can adsorb pollutants at lower concentrations (ppb), absorb pollutants that can be easily removed from the adsorbent surface, and be able to be recycled numerous times Metal oxide nanoparticle materials have been investigated recently for their potential as adsorbents due to their high adsorption capacity, low cost, simplicity of separation, and improved stability. Therefore, MONPs show promise for the industrial-scale treatment of wastewater. However, some researchers strongly believe that variations in the redox status have an impact on how dangerous these toxins are. Among the nanoparticles that are frequently used to absorb pollutants are ferric oxide, manganese oxide, zinc oxide, magnesium oxide, and titanium oxide [124]. Table 3 summarizes some applications of green MONPs for the removal of pollutants.

### 6.1. Adsorption Technique for the Elimination of Inorganic Pollutants

Fe, Hg, Cu, Ni, Cu, Zn, Cr, and Pb are harmful toxic heavy metal effluents that are released into the environment by a variety of companies, including those that perform mining operations, metal plating, and battery production (II). They do not biodegrade and tend to build up in aquatic environments. When the concentration of these metal ions exceeds the permissible level set by the WHO and the US Environmental Protection Agency, they are considered contaminants (USEPA). Due to the serious health and environmental dangers they provide, it is essential to find new and effective adsorbents and strategies to remove them from water and wastewater [26,27]. Nonetheless, few studies have been carried out on the application of green MONPs by algae for the removal of inorganic pollutants by adsorption technique.

Mahmoud et al. [125] produce green CuONPs for the removal of cadmium, nickel, and lead from contaminated wastewater. At pH 6, which simulates wastewater under typical environmental circumstances, the optimum removal efficiency of Pb(II), Ni(II), and Cd(II) was recorded to be 84%, 52.50%, and 18%, respectively, for simulating wastewater in normal environmental conditions. A green method of producing CuONPs by *Calotropis procera* was used in a study by Danish et al. [126] as an adsorbent for the quick and effective removal of Cd (II) and Fe(III) during the wastewater-treatment process. The outcomes demonstrated that the adsorption capacities for Cd(II) and Fe(III) were 84.75 and 126.32 mg/g, respectively. CuONPs were produced by Dubey and Sharma [127] to remove Cr(VI) from the aqueous solution. At initial Cr(VI) and pH values of 5 mg/L and 2.0, respectively, CuONPs were shown to have 99.99% removal efficacy, compared to 53% removal efficiency for traditionally manufactured nanoparticles. The microalga *Chlorosarcinopsis* sp. was cultivated in primary treated wastewater and combined with

ZnONPs in secondary treated wastewater for the treatment of sewage wastewater with high-quality biodiesel synthesis [104].

### 6.2. Adsorption Technique for the Removal of Organic Pollutants

Nano-adsorbents are useful for removing metal ions and organic molecules, and functionalization can improve their selectivity for particular contaminants. However, the application of green MONPs by algae for water treatment has only been covered in a few academic reviews. Mansour et al. [7] investigated green ZnONPs synthesized from *Pterocladia capillacea*, and their ability to remove an organic hazardous dye (Isomeric Violet 2R) from an aqueous solution was assessed. In addition, 0.08 g of ZnONPs was applied at a pH of 6, a temperature of 55 °C, and a contact time of 120 min, with a percentage removal of Ismate violet 2R of 99%. ZnONPs had a dye adsorption capability of 72.24 mg $g^{-1}$. Bhukal et al. [91] described the green synthesis, characterization, and highest effectiveness of nano-adsorbent iron oxide with the help of *spirulina* algae powder. $Fe_2O_3$NPs as nano-adsorbents were used for the removal of crystal violet dye from wastewater. At a pH of 12, a dose of 30 mg of nano-adsorbent, and a temperature of 35 °C, it was observed that nano-adsorbents were most effective at removing the crystal violet dye. The adsorption isotherm was most suitable using the linearized Langmuir model, with an $R^2$ value of 0.964 and a $q_{max}$ of 55.62 mg $g^{-1}$. Shalaby et al. [108] investigated the green synthesis of recyclable iron oxide nanoparticles using *Spirulina platensis* microalgae. A green iron oxide nanoparticle was applied for the removal of the cationic crystal violet (CV) and anionic methyl orange (MO) dyes from their aqueous solutions. Under a variety of optimization conditions, the effectiveness of iron oxide NPs was carefully evaluated for their adsorptive performance towards the target CV and MO dyes. Regarding the dynamic static sorption data, the pseudo-second-order model was attributed to the kinetics profile, although the Langmuir theory quantitatively dominated the sorption isotherm with maximum adsorption capacities of 256.4 mg $g^{-1}$ for CV and 270.2 mg $g^{-1}$ for MO. The results from thermodynamics confirmed that the adsorption process is endothermic. For five sorption/desorption cycles, the repeatability of the used sorbent was successfully stressed. ZnONPs and two microalgal species, *Chlorella vulgaris* and *Dunaliella tertiolecta*, eliminate hydrocarbon from oily water, although the efficacy of this removal is influenced by the type of algae, the number of hydrocarbons, and the concentration of NPs [65]. Hassan et al. [106] reported the use of iron oxide nanoparticles made by green synthesis for the elimination of pyrene and benzo (a) pyrene (PAHs) from water. The adsorption factors were observed. These variables include the dose of nanoparticles, pH, temperature, and initial PAH concentration. According to this study, the results showed that green $FeO_3$NPs have a maximal adsorption capacity of 2.8 and 0.029 mg $g^{-1}$, for both pyrene and benzo (a) pyrene, respectively. In another study, green iron nanoparticles with a spherical form and a size range of 5–10 nm were created for hydrocarbon treatment [106]. The ability of the biosynthesized iron nanoparticles to remove total petroleum hydrocarbons (TPH) from contaminated soil and water was investigated; 88.24% of TPH was removed from the water sample by the iron nanoparticles after 12 and 32 h of treatment, respectively.

### 6.3. MONPs as Adsorbents for Water Disinfection from Microbes

Algae are known to contain a wide variety of polyphenolic chemicals as primary elements of their antioxidative characteristics. The production of nanoparticles is made non-toxic and environmentally friendly by using microalgae extracts as capping and reducing agents. The effectiveness of magnetic NPs treated with polyallylamine hydrochloride (PAAH) in removing harmful germs from drinking water through electrostatic interaction and magnet capture was investigated. *Escherichia coli*, Acinetobacter, Pseudomonas, and Bacillus were the four main pathogenic species with high removal efficiencies. The study indicated high bacteria removal efficiencies (99.48%) and total bacteria residual counts as low as 78 CFU $mL^{-1}$, which exceeded the WHO drinking water criterion of 100 CFU $mL^{-1}$ [128]. Bhattacharya et al. [57] investigated how CuONPs produced from

algae can be used to disinfect drinking water. *E. coli*, a Gram-negative bacteria, was resistant to the antibacterial action. *E. coli* was successfully eliminated from surface water samples using an optimized dose of 50 g mL$^{-1}$ of CuONPs. In addition, silver oxide nanoparticles were produced using a green method using *Sargassum longifolium*. The synthesized particles exhibited antifungal efficacy against *Fusarium* sp., *S. longifolium*, *Candida albicans*, and *Aspergillus fumigatus*. CuONPs were synthesized using brown algae (*Sargassum polycystum*), and their antibacterial and anticancer properties were tested. *Pseudomonas aeruginosa* and *Aspergillus niger* both showed 96% antibacterial activity. The MCF-7 breast cell line exhibited anticancer activity, with an IC50 value of 61.25 g L$^{-1}$ [57]. Abboud et al. [80] used the brown alga *Bifurcaria bifurcata* to biosynthesize CuONPs that ranged in size from 5 to 45 nm. Likewise, it was found that these nanoparticles are highly effective antibacterial agents against *Staphylococcus aureus* (Gram-negative) and *Enterobacter aerogenes*, two different types of bacteria (Gram-positive).

## 7. Photocatalytic Applications of Algae-MONPs for Bioremediation

Recently, water pollution in surface and groundwater has become a growing concern. The presence of organic-rich waters, including dyes, chemicals, pesticides, and pharmaceuticals, has had a significant impact on the ecological balance of wastewater. Traditional wastewater treatment methods have not been able to effectively eliminate these harmful compounds. This has led to the exploration of alternative methods such as advanced oxidation processes (AOPs), which are promising for the degradation of toxic organic contaminants [129]. The use of photocatalysis, in particular, has gained attention due to its eco-friendly nature, high efficiency, cost-effectiveness, and ability to eliminate non-biodegradable pollutants [130].

ZnONPs exhibit remarkable properties in both their chemical and physical characteristics. These properties include high chemical stability and low levels of cytotoxicity. Due to these attributes, ZnONPs are ideal materials for conducting photocatalytic experiments. In this process, ZnONPs are used to catalyze chemical reactions by absorbing and utilizing light energy, making them effective and efficient catalysts for various chemical reactions. Additionally, the low toxicity of ZnONPs makes them safe for use in both research and industrial applications [106]. In a study conducted by Subhan et al. [131], *Chlorella* sp. extract was used to produce safer, more stable, and highly pure ZnONPs. This work aimed to explore the potential of ZnONPs for photosynthesis and increased photocatalytic activity toward organosulfur pollutants. This unique and environmentally friendly process, which utilizes microalgae *Chlorella* sp. extract as a natural nano-factory for the biosynthesis of ZnONPs, was used for the first time in this study. The effectiveness of the ZnONPs in catalyzing chemical reactions was tested by comparing the retention times of a dibenzothiophene (DBT) sample before and after the photocatalytic reaction with ZnONPs. The results showed that after only three hours of photocatalytic reaction, 97% of the DBT pollutant was destroyed under neutral pH conditions. This reduction in DBT concentration was identified using gas chromatography. The high efficiency and durability of the green ZnO-nano photocatalyst were also demonstrated through its rapid separation and simple recyclability over five consecutive runs. Overall, the study demonstrated the potential of *Chlorella* sp. extract as a natural nano-factory for the production of safe and highly pure ZnONPs, which have significant photocatalytic activity in breaking down organosulfur pollutants.

Table 4 shows some previous studies for the elimination of different contaminants using bioinspired synthesized MONPs by photocatalytic technique.

**Table 4.** Metal oxide-based nanomaterials for removal of organic and inorganic pollutants from wastewater by a photocatalysis process.

| Algal Species | Adsorbents (MO/MONPs) | Pollutants | Optimization Conditions | Method/Efficiency, Adsorption Capacity ($mg \cdot g^{-1}$), Removal (%) | Refs. |
|---|---|---|---|---|---|
| *Padina pavonica* | FeONPs | Congo Red | 40 min, 180 °C, pH, 4 | 95% | [132] |
| *Padina gymnospora* | CdO-ZnO | Reactive blue 198 dye | 15 min | 99.57% | [133] |
| *Chlorella pyrenoidosa* | FeONPs | Crystal violet dye | 150 min | - | [134] |
| *Sargassum wightii* | MgO | Methylene blue dye | 10 mg, 30 min | - | [135] |
| *Sargassum species* | CO-ZnO | Malachite green | 10 min, 0.05 g, pH 7 | - | [136] |
| *Sargassum species* | ZnONPs | Malachite green dye | 10 min, 0.06 g, pH 7 | - | [136] |
| *Cystoseira trinodis* | CuO | Methylene blue dye | 5 h | 43.5% | [137] |
| *Cystoseira trinodis* | CuO | Acid yellow 23 | 5 h | 66.3% | [137] |
| *Cystoseira trinodis* | CuO | Reactive black 5 | 5 h | 67.8% | [137] |
| *Ulva fasciata* | ZnO-NPs | Methylene blue (MB) | 140 min, 35 °C, pH 7 | 84.9% | [138] |
| *Chlorella vulgaris* | TiO$_2$ | Total Phosphorous | | 78% | [139] |
| *Chlorella vulgaris* | TiO$_2$/Ag | Cr(VI) | pH 2 | 91% | [140] |
| *Cystoseira trinodis* | CuO-NP | Methylene blue dye | 90 min, 25 °C, pH 2 | 92% | [82] |

Fouda et al. [138] examined the photocatalysis, tanning wastewater treatment, and characterization of psycho-synthesized ZnONPs using *Ulva fasciata*. The study found that the catalytic activity to remove the methylene blue dye increased in the presence of visible light at a concentration of 1.0 mg mL$^{-1}$ after 140 min at pH 7 and 35 °C. Compared to the dark conditions, the maximum amount of dye removed was 84.9% as compared to 53.4%. In another study, Dehghani and Mahdavi [141] used green ZnONPs and UV irradiation to remove acid dye 4092 from an aqueous solution. The dye samples were exposed to UV irradiation for 2 to 12 min. Studies revealed that the treatment of dye was achieved at 0.5 mg L$^{-1}$ of dye concentration, 12 min of radiation period, pH 5, and 0.2 g L$^{-1}$ of catalyst dose. Caf [132] investigated the biogenic manufacture of FeONPs using *Padina pavonica* algae extract applications of photocatalytic degradation of Congo red dye, neurotoxicity, and antioxidant activity. This study focused on the environmentally friendly synthesis of FeNPs by *Padina pavonica* (Pa@FeNPs) and applied it to the removal of Congo red dye. Utilizing Pa@FeNPs, the CR removal percentage attained a very high efficiency of 95%.

TiO$_2$ is one of the best photocatalysts due to its low cost, lack of toxicity, high balance, and simplicity of availability. As is typically seen, TiO$_2$ always produces the greatest photocatalytic performances with the highest quantum yields [142]. The photocatalytic mechanism of TiO$_2$ particles was described. To prevent recombination, the produced electron-hole pairs must be trapped. Adsorbed oxygen species are likely to serve as electron traps, resulting in the development of superoxide species (O$_2$), which are unstable, reactive, and capable of changing in several different ways, while hydroxyl ions (OH$^-$) are likely to act as hole traps, resulting in the formation of hydroxyl radicals, which are potent oxidants [143]. *Chlorella pyrenoidosa* has been used to synthesize green TiO$_2$NPs for the treatment of crystal violet dye by photocatalytic activity under visible light irradiation. Due to the activity of graphene oxide as a photosensitizer, its influence on this nanomaterial demonstrates a performance twice that of TiO$_2$ [144]. As reported by Wang et al. [145], Cr (IV) can be eliminated using TiO$_2$/Ag NP combined with algae with high photocatalytic efficiency when exposed to visible light. More than 91% of Cr(VI) was removed from wastewater using algae-TiO$_2$/Ag after a 3-h reaction. With varied initial concentrations of Cr (VI) at 5.0, 10.0, 30.0, 50.0, and 100.0 mg L$^{-1}$, respectively, the effect of the initial Cr (VII) concentration on the reaction was investigated in the presence of algae (TiO$_2$/AgNF) at pH 4.

Gu et al. [82] investigated the characterization and photocatalytic activity of ultrasound-assisted CuONP biosynthesis using the brown alga *Cystoseira trinodis* over 90 min at room

temperature. Furthermore, under a variety of experimental conditions, including dye concentration, catalytic load, pH, and the presence of UV and sunlight, CuONPs demonstrated significant activity as an effective photocatalyst for the degradation of methylene blue dye. The highest catalytic effect was observed at pH 4. The amount of methylene blue degradation is directly related to the amount of loaded catalyst, while its relation to dye concentration is inverse.

## 8. Conclusions

Algal (microalgae and seaweed) offers practical, non-toxic, and cost-effective biomaterials for producing different metallic nanoparticles (MONPs) with excellent energy performance. Algal biomolecules serve as a reducing and capping agent, resulting in stabilized nanoparticles. The size and shape of the aquatic plant-based MONPs depend on essential variables such as temperature, pH, extract concentration, stirring speed, and reactant ratio. Green MONPs have been widely studied for various applications, including environmental remediation. Despite promising findings, the use of algae-based MONPs for hydrocarbon pollutant recovery has been underexplored. More research is needed to investigate the fate, transport, and toxicity of MONPs in the marine environment. Algal-based MONPs such as $TiO_2$, FeO, CuO, ZnO, and MgO nanoparticles have shown significant potential in the treatment of environmental pollutants. Future applications should focus on effective techniques that employ small quantities of MONPs. Low-cost synthesis techniques and extensive testing are required to successfully deploy MONPs in the field. Optimization parameters such as pH, salt content, contact time, and temperature affect the stability of NPs with high yields, which can vary depending on the biological entities used.

## 9. Challenges and Future Prospects

Green synthesis has been utilized successfully in the last decade to synthesize a wide range of MONPs, including MONPs with varied forms and characteristics. Few research articles on the interdisciplinary routes in the synthesis of MONPs from various marine algae have been published worldwide. More research is needed to discover optimal conditions for both the biosynthesis of nanoparticles and their use in the marine environment. However, the green synthesis method has several limitations and drawbacks that limit its commercialization and subsequent applications in biotechnology and nanomedicine. The following is a summary of the present challenges:

1.  People should be aware of performance results regarding green chemistry because green technology is still developing and many items are in the research and development stage.
2.  An increase in commercialization and the number of skilled workers available to install or implement green technology-based systems or products.
3.  Policies for systems based on green technology should be finalized in the majority of nations.
4.  More study is required to develop green low-cost production techniques and to conduct extensive testing to successfully deploy metal oxide nanoparticles in the field.
5.  Before putting a nanomaterial to use in practice, it should be thoroughly researched in terms of its environmental impact, preparation costs for large-scale application, environmental policy, and ideal conditions.
6.  Stability and functionalization: more green synthesis research is needed to surface-modify synthesized MONPs to improve their stability in biological media and functionalize them with specific antibodies or peptides to better their uses in drug delivery and cancer therapy.
7.  A major challenge is the scalability of created MONPs from a laboratory method to satisfy the massive needs at industrial and pharmaceutical scales.
8.  A thorough toxicological analysis of biosynthesized MONPs is required to expand their application in various industries.

Despite the popularity of the green synthesis approach, overcoming the challenges mentioned above could improve the global acceptability and adaptability of the commercial synthesis of MONPs. Future research and development in this field should focus on overcoming current barriers and developing fresh standardized methods, constructing smart and safe MONPs functionalized with various biomolecules, and targeting moieties for multifunctional applications.

**Author Contributions:** Conceptualization, A.E.A. and M.A.; validation, A.E.A. and M.A.; project administration, M.A.; visualization, M.A. and A.T.M.; funding acquisition, M.A., A.M.A. and A.T.M.; writing—original draft preparation, M.A., A.M.A. and A.T.M.; writing—review and editing, M.A., A.M.A. and A.T.M. All authors have read and agreed to the published version of the manuscript.

**Funding:** This work was supported by the Deanship of Scientific Research, Vice Presidency for Graduate Studies and Scientific Research, King Faisal University, Saudi Arabia [GRANT3,386].

**Data Availability Statement:** Not applicable.

**Acknowledgments:** The authors would like to acknowledge the Deanship of Scientific Research, Vice Presidency for Graduate Studies and Scientific Research, King Faisal University, Saudi Arabia [GRANT3,386].

**Conflicts of Interest:** The authors declare no conflict of interest.

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
