# Peer review of "Advances in Green Synthesis of Metal Oxide Nanoparticles by Marine Algae for Wastewater Treatment by Adsorption and Photocatalysis Techniques"

_catalysts, doi:10.3390/catal13050888_

Round 1

Reviewer 1 Report

This manuscript describes the use of algae-based green synthesis of metal oxide nanoparticles for bioremediation is an environmentally friendly and cost-effective alternative to conventional approaches. The work has been carried out with care and the results have been presented with clarity and discussed appropriately. This review focuses on the biogenic fabrication of metal oxide nanoparticles based on aquatic plants. In addition, photocatalysis is highlighted as a promising tool for the remediation of industrial effluents. In this way, this manuscript demonstrates a significant contribution in the area of catalysis. Accordingly, I can recommend publication of this work in Catalysts.

Author Response

Please, find the attached file

Reviewer 2 Report

The compilation is useful and systematic.

Fig 1 ought to be corrected because the text in the central bubble is not properly divided. The 2nd sentence of Section 4.5. is misleading; the stirring rates are not given at the right positions.

Some characteristic informative figures should have been inserted.

Section 8. seems to be a bit superfluous.

Minor errors of grammar or typos occur, e.g., in the 4th paragraph of Section 1., in the last sentence of Section 2., and in the first sentence of Section 5.5.1.

Author Response

Please, find the attached file

Reviewer 3 Report

Comments to the authors:

In this review manuscript authors describe the use of algae-based green synthesis of metal oxide nanoparticles (MONPs) for bioremediation through environmentally friendly and cost-effective alternative to conventional approaches. This review article has value for the researchers in the related areas. However, the review article needs minor improvement before acceptance for publication. My detailed comments are as follow:

1.      In the introduction section authors should introduce following relevant articles like as mentioned below: doi.org/10.3390/ijms222413383, doi.org/10.1007/s40089-021-00362-w

2.      Few relevant images should be included such as results or graphical

3.      Authors should introduced a paragraph related to research challenges

4.      Write a short paragraph on future scope of such research

5.      There are few grammatical and typos errors.   

Minor editing is required.

Author Response

Please, find the attached file
